# Bayesian Spatio-Temporal Multilevel Modelling of Patient-Reported Quality of Life following Prostate Cancer Surgery

**DOI:** 10.3390/healthcare12111093

**Published:** 2024-05-26

**Authors:** Zemenu Tadesse Tessema, Getayeneh Antehunegn Tesema, Win Wah, Susannah Ahern, Nathan Papa, Jeremy Laurence Millar, Arul Earnest

**Affiliations:** 1School of Public Health and Preventive Medicine, Monash University, Melbourne, VIC 3004, Australia; 2Department of Epidemiology and Biostatistics, Institute of Public Health, College of Medicine and Health Sciences, University of Gondar, Gondar P.O. Box 196, Ethiopia; 3Garvan Institute of Medical Research, Darlinghurst, NSW 2010, Australia; 4Department of Radiation Oncology, Alfred Health, Melbourne, VIC 3004, Australia

**Keywords:** Bayesian, multilevel, spatio-temporal, quality of life, prostate cancer, Victoria

## Abstract

Background: Globally, prostate cancer is the second leading cause of cancer deaths among males. It is the most commonly diagnosed cancer in Australia. The quality of life of prostate cancer patients is poorer when compared to the general population due to the disease itself and its related complications. However, there is limited research on the geographic pattern of quality of life and its risk factors in Victoria. Therefore, an examination of the spatio-temporal pattern and risk factors of poor quality of life, along with the impact of spatial weight matrices on estimates and model performance, was conducted. Method: A retrospective study was undertaken based on the Prostate Cancer Outcome Registry—Victoria data. Patient data (*n* = 5238) were extracted from the Prostate Cancer Outcome Registry, a population-based clinical quality outcome assessment from 2015 to 2021. A Bayesian spatio-temporal multilevel model was fitted to identify risk factors for poor quality of life. This study also evaluated the impact of distance- and adjacency-based spatial weight matrices. Model convergence was assessed using Gelman–Rubin statistical plots, and model comparison was based on the Watanabe–Akaike Information Criterion. Results: A total of 1906 (36.38%) prostate cancer patients who had undergone surgery experienced poor quality of life in our study. Belonging to the age group between 76 and 85 years (adjusted odds ratio (AOR) = 2.90, 95% credible interval (CrI): 1.39, 2.08), having a prostate-specific antigen level between 10.1 and 20.0 (AOR = 1.33, 95% CrI: 1.12, 1.58), and being treated in a public hospital (AOR = 1.35, 95% CrI: 1.17, 1.53) were significantly associated with higher odds of poor quality of life. Conversely, residing in highly accessible areas (AOR = 0.60, 95% CrI: 0.38, 0.94) was significantly associated with lower odds of poor prostate-specific antigen levels. Variations in estimates and model performance were observed depending on the choice of spatial weight matrices. Conclusion: Belonging to an older age group, having a high prostate-specific antigen level, receiving treatment in public hospitals, and remoteness were statistically significant factors linked to poor quality of life. Substantial spatio-temporal variations in poor quality of life were observed in Victoria across local government areas. The distance-based weight matrix performed better than the adjacency-based matrix. This research finding highlights the need to reduce geographical disparities in quality of life. The statistical methods developed in this study may also be useful to apply to other population-based clinical registry settings.

## 1. Introduction

Globally, prostate cancer is the fourth most commonly diagnosed cancer, and an estimated 1.5 million prostate cancer cases were observed in 2020 [1]. In Australia, prostate cancer is the most commonly diagnosed cancer and the second leading cause of cancer-related deaths among men [2]. It is estimated that 18,110 new cases of prostate cancer are diagnosed every year [3]. The typical method for treating localized prostate cancer includes the surgical removal of the prostate gland, called prostatectomy [4]. Patients who undergo prostatectomy are at increased risk of sexual dysfunction, bowel dysfunction, urinary irritation, urinary incontinence, and impaired hormonal functions secondary to treatment toxicities [5]. The impact of these functional outcomes on patient quality of life (QoL) can be mitigated through early diagnosis [6]. The QoL of prostate cancer patients with prostatectomy is highly compromised because of the treatment and associated complications. While prostatectomy proves effective in the management of prostate cancer, patients who have had prostatectomy are at higher risk of a poorer QoL [7]. 

Based on the available evidence, QoL among prostate cancer patients has been found to be associated with factors like access to specialized nurses [8], geographic factors such as remoteness and relative socio-economic disadvantages [9,10], geographic variation [11], income, and comorbidity [12]. According to the Prostate Cancer Registry report, the QoL domains exhibit variation across areas and over time. Examining the spatial distribution of the experience of prostate cancer patients related to QoL is crucial to ensure effective interventions and resource allocation and for policy and advocacy [13]. Therefore, we examined the individual- and area-level determinants of QoL among prostate cancer patients who underwent surgery using the Prostate Cancer Outcome Registry—Victoria (PCOR-Vic) data.

The PCOR-Vic provides high-quality data across geography and time [14], making it suitable for the application of advanced statistical approaches such as the Bayesian spatio-temporal multilevel model. We applied a multilevel model called hierarchical related regressions, wherein both area- and individual-level factors were incorporated [15,16]. The model accounts for the spatial and temporal dimensions in studying health outcomes [17]. 

In Bayesian spatio-temporal multilevel models [18], a common choice of prior distribution for structured spatial random effects is the conditional autoregressive (CAR) model [19]. This model uses different spatial weight matrices to quantify its influence on estimates and model performance [20,21]. Spatial weight matrices enable us to incorporate spatial relationships among adjacent areas (local government areas (LGAs) based on their location, proximities, and common boundaries [22]. Our study objective is two-fold: (1) to identify area- and individual-level factors associated with poor QoL among prostate cancer surgery patients and (2) to quantify the effect of spatial weight matrices on the estimates and model performance. 

## 2. Methods 

### 2.1. Data Source

Victoria, the second most populous state in Australia, has an estimated population size of 6.68 million out of a total population of 25.69 million [23] in Australia. The PCOR-Vic dataset currently captures 90% of all newly diagnosed cases of prostate cancer in Victoria. The PCOR-Vic collects data on treatment, diagnostic, demographic, and QoL indicators 12 months after treatment [14]. The prostate cancer patients’ QoL indicators are measured by the Expanded Prostate Cancer Index Composite-26 (EPIC-26) approximately 12 months after treatment. All prostate cancer patients who underwent surgery and enrolled in the PCOR-Vic between 2015 and 2021 were included in this study. The PCOR-Vic includes opt-out consent for patients recruited into the registry. The PCOR-Vic administers the validated PCOR instrument EPIC-26 [24]. The three main methods of completing the questionnaire are telephone, email, and mail for self-completion. Victoria has a total of 698 postcodes and 80 LGAs. For this study, LGAs are used as a spatial unit of analysis. The data used for this study are stored in the Monash Secure Research Platform (SeRP) system. The analysis was performed within the SeRP system. 

### 2.2. Inclusion and Exclusion Criteria 

Prostate cancer patients who did not undergo surgery, had data recorded before the initiation of EPIC-26, and postcodes located outside Victoria were excluded from our study. Finally, 5238 patients were included in the final analysis (Figure 1). 

### 2.3. Variables 

#### 2.3.1. Outcome Measure (QoL Assessment)

Five separate domains of QoL, such as sexual function, bowel function, urinary irritate, urinary incontinence, and hormonal functions were measured by EPIC-26 among prostate cancer patients who had undergone surgery at 12-month post treatment. As the EPIC-26 was initiated in 2015, data collected from 2015 to 2021 were considered for this study. Each domain has a different number of questions to assess their quality of life. Urinary incontinence has 3 questions, urinary irritation has 5 questions, bowel function has 5 questions, sexual function has 4 questions, and hormonal function has 5 questions. A score out of 100 was given for each domain. After scoring, data were categorized into quartiles for each domain 1 (worst) to 4 (best) as data were skewed. All domains were also added together to create a composite score. The sum of these values gives a derived score ranging from 5 (worst) to 20 (best). Patients who scored below the median level were defined as having poor QoL, while those who scored median and above level had good QoL [10].

#### 2.3.2. Independent Variables

##### Patient-Level Variables

De-identified patient-level data were extracted from the PCOR-Vic registry between 2015 and 2021. Patient-level data included age (≤55, 56–65, 66–75, and 76–85), date of surgery, National Comprehensive Cancer Network (NCCN) group (low risk, intermediate risk, high risk, very high risk/metastatic), International Society of Urological Pathologists group (ISUP1, ISUP2, ISUP3, ISUP4/ISUP5), PSA (≤10, 10.01–20, >20) level, clinical T stage (T1, T2, T3/T4, unknown), treating institution type (private, public), and treating hospital location (metro, regional) were used for this study.

##### Area-Level Variables

The Index of Relative Socio-economic Disadvantage (IRSD) was obtained from the Australian Bureau of Statistics (ABS) [25], as referenced in the literature [9,17]. A higher IRSD score for a local government area (LGA) indicates a greater proportion of relatively advantaged residents. We categorized the IRSD into quartiles: 1st, 2nd, 3rd, and 4th. Individuals residing within the 25% most disadvantaged areas were assigned a quartile one, while those in the 25% most advantaged areas were assigned a quartile 4. 

The objective evaluation of relative geographical remoteness involves utilizing the Accessibility/Remoteness Index of Areas Plus (ARIA+). ARIA+ was created by the Department of Health and Aged Care of the Australian Government to gauge accessibility at the local government area (LGA) level. It comprises five classifications: extremely remote (ARIA+ score > 9.08), remote (ARIA+ score 5.80–9.08), moderately accessible (ARIA+ score 3.51–5.80), accessible (ARIA+ score 1.84–3.51), and highly accessible (ARIA+ score < 1.84) [26]. This score was calculated based on factors such as access to goods and services, distance from towns and cities, and opportunities for social interactions. The geographic information for patients was represented by their postcode, and we established correspondence to LGA using ArcGIS through the “intersection” technique [27]. We chose LGAs as the unit of analysis due to the accessibility of census data from the Australian Bureau of Statistics (ABS), and it is the preferred unit for targeting policy interventions effectively.

### 2.4. Statistical Analysis 

We employed the hierarchical related regression (HRR) approach, using Bayesian spatio-temporal multilevel models [15,16]. This modelling strategy allows for the integration of individual- and area-level covariates with the outcome by accounting spatially structured, unstructured and non-linear trends into a unified and parsimonious model. The HRR model has been modified to consider spatial dependency by specifying a Besag, York and Mollie (BYM) model [28] and quadratic temporal trends [29]. Additional detailed information about the model can be found in Appendix A. 

We initiated two Markov Chain Monte Carlo (MCMC) chains to draw inferences from the posterior distribution, each beginning with distinct starting values and same priors [30]. For fixed-effects individual-level covariates and random-effects area-level covariates, we specified Normal priors (mean = 0, precision = 0.725). This assumes a 95% probability that the true odds ratio (OR) was within the range of 0.1–10 [16].

Two different models were fitted using different adjacency matrices. The first model employed Queen’s adjacency-based spatial weight matrices. In this technique, units are considered neighbours if they share a common boundary point or vertex without necessitating alignment along entire edges [18]. The second model was fitted by specifying distance-based spatial weight matrices. In distance-based spatial weight matrices, neighbourhood relations are expressed as a function of the distance between each LGA. For this study, a distance band of 131 km was used, and this distance band selected all LGAs with at least one neighbour. Additional detailed information about the spatial weight matrices can be found in Appendix A. Model convergence was assessed using Gelman–Rubin statistic plots (Appendix A). For model selection, the WAIC was used. Lower WAIC values represent a better fit and the importance of covariates. In both models, we began with the most significant covariate identified through univariate results. Subsequently, we sequentially added additional covariates to evaluate their impact on improving model fit, as measured by the WAIC [31]. 

The model was estimated using MCMC algorithms with two chains with a different initial values and same priors. There were a total of 100,000 MCMC iterations run with a burn-in of 50,000 and thinning set to 2. Model convergence was examined through visual inspections of MCMC chains, and the Gelman–Rubin statistic plots and model selection were performed based on the WAIC. We presented the adjusted odds ratio (AOR) and 95% credible intervals (CrI) for the posterior estimates and mapped the relative risks (RRs) of poor QoL areas in Victoria by dividing observed counts by the expected counts for each LGA and by year of surgery.

The relative contribution of spatial variation was calculated by dividing the variance of structured random-effect var(u) by the sum of the variance of structure var(u) and unstructured random-effect var(v) using different weight matrices, empirically represented as ∅ = (var(u))/(var(u) + var(v)). Values ∅ close to 1 indicate that the structured random variation effect dominates, while values ∅ close to 0 mean that unstructured random-effect variation dominates. The effect of using different spatial weight matrices on the coefficient estimates was calculated and presented in the regression table.

STATA version 17 and MultiBUGS version 2 statistical software were employed for data extraction, management, and analysis. The spatial weight matrices were generated using GeoDA version 1.22, and Arc-GIS version 10.8 software was utilized for creating all maps.

## 3. Results 

### 3.1. Descriptive Statistics

A total of 5238 prostate cancer patients who underwent surgery between 2015 and 2021 were included in this study. Patients who completed EPIC-26 questionnaire 12-month post-treatment were included in this study. The median age of this study participants was 66 years with IQR ± 9. Of 5238 patients, 3696 (70.56%) resided in the metropolitan treating hospital location. The majority of the patients (77.47%) had a less than 10 PSA level score at diagnosis (Table 1). 

### 3.2. Correlation of QoL with Covariates 

The QoL outcome was assessed across five domains. Of the total 5238 prostate cancer patients who had undergone surgery, 1906 (36.39%) of them had poor QoL after 12 months of treatment. The chi-square analysis showed that QoL had a significant association with a list of covariates, as presented below, with a *p*-value less than 0.001 (Table 2). 

### 3.3. Mapping of Relative Risk of Poor QoL

We mapped the relative risks of poor QoL among prostate cancer patients who had surgery. We observed spatial and temporal variation in the relative risks across Victoria. In certain LGAs, the actual observed number of cases exceeded the expected values (RR > 1), indicating areas with a high prevalence of poor QoL in that LGA. The relative risks of poor QoL exhibited variation across LGAs and over time in Victoria, ranging from 0 to 2.23 between 2015 and 2021. These maps effectively illustrate areas with high and low risks for poor quality of life. The high dense colour indicates high levels of poor quality of life, and the less dense colour indicates a better quality of life. This information provides geographic-based evidence for effective intervention and designing strategies to increase QoL among prostate cancer surgery patients (Figure 2).

### 3.4. Model Development 

This study examined a total of nine variables, encompassing seven individual-level factors, age group, NCCN group, Gleason risk group, PSA level, clinical T stage, institution, and hospital, and two area-level factors, namely IRSD and remoteness. Both univariable and multivariable analysis were conducted. Initially, univariable analyses were performed by considering one variable at a time. The univariable analysis revealed that variables such as age group, NCCN group, PSA level, institution, and accessibility had at least one significant category. In the development of the multivariable model, each variable was included sequentially, starting with the most important one, and we assessed the impact on model performance by examining the WAIC (Table 3).

### 3.5. Model Selection and Comparison

The model was estimated using MCMC algorithms with two chains with different initial values and priors. A total of 100,000 MCMC iterations run with a burn-in 50,000 and thinning set to 2. Both distance-based and adjacency-based spatial weight matrices were utilized for smoothing. We developed Bayesian spatio-temporal multilevel models based on these two adjacencies and compared them using the WAIC. The findings indicate that the model relying on a distance-based matrix performs better as compared to the adjacency-based matrix (Table 4).

### 3.6. Model Convergence

For model convergence, the Brooks–Gelman–Rubin statistics plot was used. Convergence is assumed when the potential scale reduction factor values approach 1. Based on the results, we found that the model converged (Appendix A).

### 3.7. Risk Factors of Poor QoL

In the multivariable analysis, age group, PSA level, treating institution and accessibility were found to be significantly associated with poor QoL among prostate cancer surgery patients in Victoria. Patients aged 66–75 years and 76–85 years had 1.70-fold (AOR = 1.70, 95% CrI: 1.39, 2.08), and 2.90-fold (AOR = 2.90, 95% CrI: 1.39, 2.08) higher odds of poor QoL compared those aged under or equal 55, respectively. Patients with PSA levels of 10.1–20.0 had 1.33-fold (AOR = 1.33, 95%CrI: 1.12, 1.58) higher odds of poor QoL as compared to PSA levels less than 10. The odds of poor QoL among prostate cancer patients treated in public hospitals increased by 35% (AOR = 1.35, 95% CrI: 1.17, 1.53) compared to prostate patients treated in private hospital. ARIA+ was found to be significantly associated with poor QoL among prostate cancer surgery patients. The odds of poor QoL among prostate cancer patients residing in highly accessible area were decreased by 40% (AOR = 0.60, 95% CrI: 0.38, 0.94) compared to low accessible areas. We have also quantified the effect of using distance versus adjacency-based spatial weight matrices in the Bayesian spatio-temporal multilevel model. Higher estimated percentage changes in the coefficients were observed in the age group category from 66 to 75 (4.8%) and PSA level from 10.1 to 20.0 (4.5%) (Table 5).

### 3.8. Relative Contribution of Spatial Structured and Unstructured Random Effects

Using a distance-based spatial weight matrix at the LGA level, the structured random effect accounted for 78% of the total spatial variation in poor QoL among prostate cancer patients who had surgery. In contrast, using an adjacency-based spatial weight matrix at the LGA level accounted for 76% of the total spatial variation in poor QoL among these patients. (Table 6).

### 3.9. Spatial Weight Matrices

We considered both distance- and adjacency-based spatial weight matrices to quantify the relative influence of the choice of these weights on model performance and estimates. In the adjacency-based weight matrix (Queen-I), the minimum neighbour was 1 and the maximum neighbour was 9, while in the distance-based weight matrix, the minimum to maximum was 1 to 48 (Table 7). The distance-based weight matrix considers a large number of neighbours based on the given distance band (131 km) as compared to the adjacency-based weight matrix, which considers neighbours if they only share a common vertex of LGA.

## 4. Discussion

### 4.1. Interpretation of the Results

In response to the limited evidence regarding the spatio-temporal patterns and factors linked to poor QoL among prostate cancer patients who underwent surgery in Victoria, we found several important individual- and area-level risk factors of poor QoL. In our model, the spatial autocorrelation was modelled by the CAR prior distribution. In the CAR model, the spatial weight matrix has a prominent role in determining the degree of spatial smoothing. Therefore, in addition to identifying the risk factors, this study quantifies the impact of choice of spatial weight matrices on both estimates and model performance. In the multivariable Bayesian spatio-temporal multilevel conditional autoregressive model, age group, PSA level, institution and accessibility were found to be significantly associated with poor QoL among prostate cancer patients who underwent surgery.

We mapped the relative risks of poor QoL among prostate cancer patients who underwent surgery, revealing significant spatial and temporal variations in poor QoL in Victoria. Additionally, we evaluated the effect of spatial weight matrices on estimates and model performance, and we found considerable differences based on types of spatial weight matrices used.

Age was significantly associated with poor QoL. As age increased, the odds of poor QoL also increased. This could be due to the fact that as age increases, the risk of urinary incontinence, urinary bother, and sexual dysfunction also increase [32]. This finding is in line with a study conducted by Kurian et al. [33], which examined age and QoL among prostate cancer patients reported that old age negatively affected health-related quality of life. The study performed by Popiołek et al. [34] also found advanced age reduced QoL among prostate cancer surgery patients.

PSA levels have a significant role as a biomarker in the management of prostate cancer. From our study, we found that PSA level was significantly associated with poor QoL among prostate cancer patients who underwent surgery. This is also supported by previous studies [35,36]. This increase may be attributable to the routine practice of monitoring PSA levels after surgery [37]. A rise in PSA levels following surgery could signal the presence of residual cancer cells or recurrence, potentially impacting the patient’s QoL as concerns arise regarding the efficacy of the treatment [38].

Our study revealed that treating institution was a significant factor for poor QoL among prostate cancer surgery patients. The odds of poor QoL among prostate cancer patients treated in public hospitals increased by 35% as compared to private institutions. This finding is supported by the previous literature [39,40]. This might be due to the fact that private institutions often had better resources and facilities and provided better access to patients compared to the public. Additionally, private institutions may have better skilled healthcare professionals, advanced technology, and a range of supportive services, which may result in differences in QoL among prostate cancer surgery patients [41].

ARIA+ was found to be significantly associated with poor QoL among prostate cancer patients who underwent surgery. The odds of poor QoL among prostate cancer surgery patients residing in highly accessible areas was lower. Our finding is in agreement with the previous literature [10,17,41,42], which could be due to the fact that remote areas are affected by different factors like distance to a healthcare facility, inaccessibility of infrastructure, the difficulty of accessing specialized healthcare professionals and varying postoperative care, which results in difference in QoL among prostate cancer surgery patients [43]. Furthermore, community social interaction is also influenced by remoteness. Social support is a key intervention for patients to increase their QoL, but remote areas commonly have less social interaction, which may result in a diminished QoL [44].

According to the results of our study, there was substantial spatial and temporal variation in poor QoL among prostate cancer surgery patients in Victoria at the LGA level from 2015 to 2021 l. One possible explanation for this might be variability in accessibility of healthcare services across different areas of Victoria, which might contribute to differences in poor quality of life. Patients in areas with limited access to specialized care, mental health services, or rehabilitation support might experience more difficulties. In addition, COVID-19 may also have a separate effect on this variation. This finding is supported by a systematic review performed by Dasgupta et al. [45].

The secondary objective of this study was to evaluate and measure the effect of spatial weight matrices on estimate and model performance. We fitted the Bayesian spatio-temporal multilevel conditional autoregressive model using both distance- and adjacency-based spatial weight matrices. We observed considerable differences in the estimates and model performance. We compared both types of spatial weight matrices using the WAIC. The distance-based weight spatial weight matrix performed better than the adjacency-based. The finding is consistent with studies performed by Earnest et al. [18], who evaluated four adjacency-based matrices and seven distance-based spatial weight matrices, ultimately concluding that distance-based spatial weight matrices exhibited superior performance and a systematic review conducted by Duncan et al. [22] also found that the distance-based spatial weight matrix performs relatively well. This might be due to the fact that the distance-based weight matrix considers a larger number of neighbours while the adjacency-based weight matrix only takes its immediate next LGA as a neighbour if the LGA is adjacent. We found a considerable percentage change in estimate for variables like PSA and age when fitting models using the adjacency-based spatial weight matrix and the distance-based spatial weight matrix. Effect size becomes bigger for distance-based weights due to greater smoothing by distance-based weights and the nature of spatial variation for covariates [46].

### 4.2. Strengths and Limitations of This Study

The strengths of this study include the following: (1) Bayesian spatio-temporal multilevel modelling which has clinical application novelty—this model enables us to assess the joint and independent effects of individual- and area-level covariates that explain variation in individual- and area-level outcomes, which is a single, parsimonious model resulting in reduced ecological bias and improved precision; (2) the model also considers time, spatially structured and unstructured random effects, hence accounting for contextual effects. Furthermore, this advanced statistical model was implemented based on a population-based multicenter large registry that provides a large sample size and improved precision.

The following limitations should be taken into consideration when interpreting this study’s findings. Because this study was based on retrospective data, we are unable to infer cause-and-effect relationships. Furthermore, only 90% of prostate cancer cases in Victoria are covered by the registry, which may limit the generalizability of the results to all Victoria cancer patients. Other uncontrolled factors like physician–patient communication, patients’ beliefs and preferences, and individual-level socio-economic status were not available, which could potentially affect our results.

## 5. Conclusions

The prevalence of overall poor QoL was high (36.39%) among prostate cancer surgery patients in Victoria from 2015 to 2021. Older age group, high PSA level, treating institution and remoteness were statistically significant factors associated with poor quality of life. We mapped relative risks associated with poor QoL and the result showed that there was spatial and temporal variation in poor QoL in Victoria. We evaluated and quantified the effect of spatial weight matrices and we found considerable differences in estimate and model performance. The result showed that the distance-based weight matrix performed better than the adjacency-based weight matrix. Low accessible areas and old age lead to a poorer quality of life, highlighting the need for geographically targeted treatment to reduce disparity in QoL among prostate cancer surgery patients in Victoria. Researchers should also perform a sensitivity analysis as to the choice of neighbourhood weight matrices. The Bayesian spatio-temporal multilevel model provides a robust framework for analyzing QoL data and can be applied to other population-based clinical registry settings.

## Figures and Tables

**Figure 1 healthcare-12-01093-f001:**
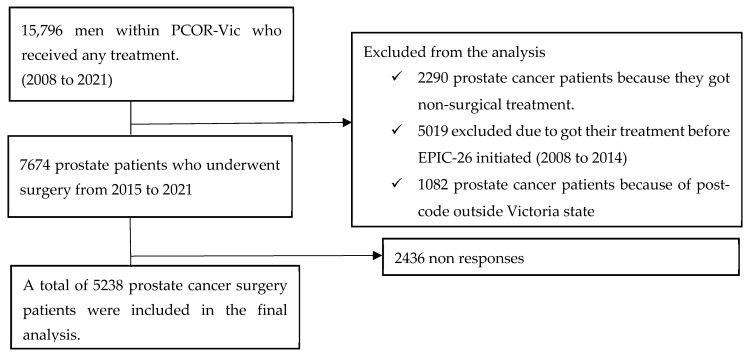
Inclusion and exclusion criteria of this study.

**Figure 2 healthcare-12-01093-f002:**
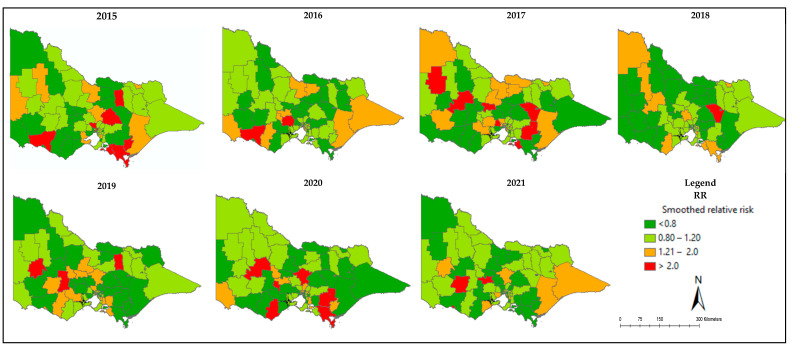
Relative risk of poor QoL among prostate cancer surgery patients in Victoria from 2015 to 2021.

**Table 1 healthcare-12-01093-t001:** Characteristics of the study population.

Variables	Category	n (%)
Age	Median (IQR)	66(9)
Age group	≤55	590 (11.26%)
	56–65	2110 (40.28%)
	66–75	2376 (45.36%)
	76–85	162 (3.09%)
Year of surgery	2015	749 (14.30%)
	2016	860 (16.42%)
	2017	914 (17.45%)
	2018	872 (16.60%)
	2019	805 (15.37%)
	2020	726 (13.86%)
	2021	312 (5.96%)
Treating hospital location	Metro	3696 (70.56%)
	Regional	1542 (29.44%)
Treating institution	Public	1423 (27.21%)
	Private	3813 (72.79%)
Gleason score at diagnosis	≤6	481 (9.31%)
	7	3700 (71.58%)
	8	554 (10.72%)
	9	424 (8.20%)
	10	9 (0.17%)
ISUP *	1	457 (8.72%)
	2	2483 (47.40%)
	3	1151 (21.97%)
	4	721 (13.76%)
	5	426 (8.13%)
PSA * at diagnosis	≤10	4058 (77.47%)
	10.1–20.0	719 (13.73%)
	>20	461 (8.80%)
National Comprehensive Cancer Network (NCCN) risk group	Low risk	294 (5.61%)
	Intermediate risk	3491 (66.65%)
	High risk	1035 (19.76%)
	Very high risk/Metastatic	418 (7.98%)
Clinical T stage	T1	2167 (41.37%)
	T2	1279 (24.42%)
	T3/T4	200 (69.61%)
	Unknown	1592 (30.39%)
Area-level variables		
IRSD *	Quartile 1	20 (25.32%)
	Quartile 2	22 (27.85%)
	Quartile 3	18 (22.78%)
	Quartile 4	19 (24.05%)
Remoteness	Accessible	18 (22.78%)
	Moderately accessible	7 (8.86%)
	Highly accessible	54 (68.35%)

* IRSD: Index of Relative Socio-Economic Disadvantage, ISUP: International Society of Urological Pathologists, and PSA: prostate-specific antigen.

**Table 2 healthcare-12-01093-t002:** Descriptive statistics of characteristics of prostate cancer patients by QoL.

Variables	QoL		Total	*p*-Value
	Poor Quality of Life	Good Quality of Life		
Quality of life	1906 (36.39%)	3332 (63.61%)	5238	
Age group				
≤55	138 (23.39%)	452 (76.61%)	590	<0.001
56–65	675 (31.39%)	1435 (68.01%)	2110	
66–75	1005 (42.30%)	1371 (57.70%)	2376	
76–85	88 (54.32%)	74 (45.68%)	162	
Treating hospital location				
Metro	1309 (35.42%)	2378 (64.58%)	3696	0.024
Regional	597 (38.72%)	954 (61.28%)	1542	
Treating institution				
Public	626 (43.93%)	799 (56.07%)	1425	<0.001
Private	1280 (33.57%)	2533 (66.43%)	3813	
Gleason score at diagnosis				
6 or less	134 (29.32%)	323 (70.68%)	454	<0.001
7	1236 (34.01%)	2398 (65.99%)	3634	
8	246 (45.39%)	296 (54.61%)	542	
9	221 (58.87%)	197 (47.13%)	418	
10	5 (62.05%)	3 (37.50%)	8	
Gleason risk group				
ISUP1	134 (29.32%)	323 (70.68%)	457	<0.001
ISUP2	809 (32.58%)	1674 (67.42%)	2483	
ISUP3	427 (37.10%)	724 (62.90%)	1151	
ISUP4/ISUP5	536 (46.73%)	611 (46.73%)	1147	
PSA * at diagnosis				
≤10	1384 (34.11%)	2674 (65.89%)	4058	<0.001
10.1–20.0	333 (46.31%)	386 (53.69%)	719	
>20	189 (41.00%)	272 (59.00%)	461	
NCCN * group				
Low risk	74 (25.17%)	220 (74.83%)	294	<0.001
Intermediate risk	1165 (33.37%)	2326 (66.63%)	3491	
High risk	476 (45.99%)	559 (54.01%)	1035	
Very high risk/Metastatic	191 (45.69%)	227 (54.31%)	418	
T stage				
T1	700 (32.30%)	1467 (58.23%)	2167	<0.001
T2	518 (40.50%)	761 (59.50%)	1279	
T3/T4	96 (46.50%)	107 (53.50%)	200	
Unknown	595 (37.37%)	997 (62.63%)	1592	

* = NCCN: National Comprehensive Cancer Network, ISUP: International Society of Urological, and PSA: prostate-specific antigen.

**Table 3 healthcare-12-01093-t003:** Model fit for poor QoL among prostate cancer surgery patients.

Model	WAIC
$\alpha +v_{j}+u_{j}+t+t^{2}$ +β1age group	11,740
$\alpha +v_{j}+u_{j}+t+t^{2}$ + β1age group+ β2nccn	11,731
$\alpha +v_{j}+u_{j}+t+t^{2}$ + β1age group+ β2nccn+ β3PSA	11,721
$\alpha +v_{j}+u_{j}+t+t^{2}$ + β1age group+ β2nccn+ β3PSA +β4institution	11,709
$\alpha +v_{j}+u_{j}+t+t^{2}$ + β1age group+ β2nccn+ β3PSA +β4institution + β5accessibility	11,658
Full model WAIC	11,658

Note: α: intercept, *u_j_*: spatially structured random effect, *v_j_*: unstructured random effect, *t* and *t*^2^: temporal component, β_1…5_: regression parameters, nccn: National Comprehensive Cancer Network, and PSA: prostate-specific antigen.

**Table 4 healthcare-12-01093-t004:** Model comparison based on distance-based and adjacency-based weight matrices.

Model	WAIC
Adjacency-based weight matrix (Queen-1)	13,110
Distance-based weight matrix	11,658

WAIC: Watanabe–Akaike Information Criteria.

**Table 5 healthcare-12-01093-t005:** Unadjusted and adjusted odds ratio estimate of poor QoL among prostate cancer surgery patients.

Variables	Adjacency-Based Weight Matrix (Queen-1)		Distance-Based Weight Matrix		Estimated % Change from Adjacency-Based Weight Matrices
	OR (95% CrI)	AOR (95% CrI)	OR with 95% CrI	AOR 95% CrI	
Age group					
≤55	1	1	1	1	
56–65	0.70 (0.62, 1.01)	0.68 (0.59, 1.13)	1.09 (0.90, 1.34)	1.08 (0.89, 1.33)	1.1
66–75	1.12 (1.01, 1.26) *	1.10 (1.07, 1.12) *	1.71 (1.40, 2.09) *	1.70 (1.39, 2.08) *	4.5
76–85	2.06 (1.48, 2.88) *	2.04 (1.46, 2.86) *	2.91 (2.01, 4.26) *	2.90 (2.00, 4.25) *	0.49
National Comprehensive Cancer Network (NCCN) Risk Group					
Low risk	1	1	1	1	
Intermediate risk	0.64 (0.55, 1.01)	0.62 (0.53, 1.01)	0.69 (0.47, 1.02)	0.68 (0.46, 1.02)	0.19
High risk	1.07 (0.90, 1.27)	1.05 (0.88, 1.25)	0.99 (0.63, 1.58)	0.97 (0.61, 1.57)	1.62
Very high risk/Metastatic	1.20 (1.02, 1.54) *	1.18 (1.00, 1.52) *	1.23 (1.01, 2.07) *	1.22 (0.72, 2.06)	0.20
Gleason Risk Group					
ISUP1	1		1		
ISUP2	0.94 (0.68, 1.30)		0.98 (0.71, 1.34)		
ISUP3	1.03 (0.73, 1.44)		1.06 (0.76, 1.47)		
ISUP4/ISUP5	1.00 (0.65, 1.55)		1.07 (0.71, 1.56)		
Clinical T Stage					
T1	1		1		
T2	0.84 (0.69, 1.00)		1.13 (0.99, 1.32)		
T3/T4	1.24 (0.90, 1.69)		1.05 (0.75, 1.47)		
Unknown	1.01 (0.87, 1.18)		1.06 (0.91, 1.22)		
Prostate-Specific Antigen (PSA)					
≤10	1	1	1	1	
10.1–20.0	1.07 (1.02, 1.45) *	1.05 (1.01, 1.43) *	1.34 (1.13, 1.59) *	1.33 (1.12, 1.58) *	4.8
>20	1.01 (0.69, 1.51)	0.99 (0.67, 1.49)	0.88 (0.67, 1.15)	0.87 (0.65, 1.13)	1.28
Treating Institution					
Private	1	1	1	1	
Public	1.35 (1.18, 1.57) *	1.33 (1.16, 1.55) *	1.36 (1.18, 1.56) *	1.35 (1.17, 1.53) *	0.05
Treating Hospital Location					
Regional	1		1		
Metro	0.81 (0.58, 1.01)		0.92 (0.75, 1.10)		
Index of Relative Socio-economic Disadvantage (IRSD)					
1st quartile	1		1		
2nd quartile	0.35 (0.10, 1.12)		1.16 (0.81, 1.66)		
3rd quartile	0.47 (0.12, 1.63)		1.06 (0.55, 1.76)		
4th quartile	0.93 (0.28, 2.93)		1.08 (0.51, 3.45)		
Accessibility/Remoteness Index of Areas Plus (ARIA+)					
Accessible	1	1	1	1	
Moderately accessible	0.64 (0.41, 1.13)	0.62 (0.39, 1.11)	0.54 (0.35, 0.97) *	0.53 (0.34, 1.01)	0.32
Highly accessible	0.49 (0.38, 0.67) *	0.47 (0.36, 0.65) *	0.61 (0.39, 0.96) *	0.60 (0.38, 0.94) *	0.32

* CrI: credible interval, OR: odds ratio, and AOR: adjusted odds ratio.

**Table 6 healthcare-12-01093-t006:** Relative structured random-effect variation using distance-based and adjacency-based weight matrices.

Parameters	Posterior Mean with 95% Credible Interval	
	Distance-Based Weight Matrices	Adjacency-Based Weight Matrices
var(u)	0.807	0.701
var(v)	0.223	0.213
∅	0.78	0.76

Note: var(u): variance of spatially structured random effect, var(v): variance of unstructured random effect, and ∅: relative spatial variation in spatially structured random effect.

**Table 7 healthcare-12-01093-t007:** Characteristics of neighbourhood weight matrices showing the number of neighbours per LGA.

Neighbourhood Type	Mean	Median	Min	Max	SD
Adjacency-based weight matrix (Queen-I)	5.11	5	1	9	2
Distance-based weight matrix	30	40	1	48	21.94

## Data Availability

Restrictions apply to the availability of these data. Data were obtained from the Prostate Cancer Outcome Registry (PCOR-Vic) and are available with the permission of the Prostate Cancer Outcome Registry.

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
