# Peer review of "Bayesian Spatio-Temporal Multilevel Modelling of Patient-Reported Quality of Life following Prostate Cancer Surgery"

_healthcare, 2024, doi:10.3390/healthcare12111093_

Round 1

Reviewer 1 Report

Comments and Suggestions for Authors

Overall, this is a polished, well written paper addressing an important health condition in Australia.  The background is well researched and presented, the objectives are clear, and the methodology is appropriate and familiar to those in the field of research.  I was hoping from the word “multilevel” in the title that this would extend to the spatial hierarchy too, with some finer resolution spatial variables.  While I understand the authors’ choice to use LGAs as stated on page 4, I think it is possible to acquire estimates of remoteness and socioeconomic status at a smaller geography, e.g. SA2.  Modelling at the SA2 level might provide additional insights, and aggregation back to LGAs can be used for policy interventions.  At the least, this should be considered as a potential future extension.

Minor remark: P8,figure 2: I would suggest using a diverging colour scheme (different colours for high and low ends of the RR spectrum) to better delineate regions of low and high RR at a glance.

Author Response

Reviewer #1

Overall, this is a polished, well written paper addressing an important health condition in Australia.  The background is well researched and presented, the objectives are clear, and the methodology is appropriate and familiar to those in the field of research.

 I was hoping from the word “multilevel” in the title that this would extend to the spatial hierarchy too, with some finer resolution spatial variables.  While I understand the authors’ choice to use LGAs as stated on page 4, I think it is possible to acquire estimates of remoteness and socioeconomic status at a smaller geography, e.g. SA2.  Modelling at the SA2 level might provide additional insights, and aggregation back to LGAs can be used for policy interventions.  At the least, this should be considered as a potential future extension.

Author’s response: Thank you for your comments. We chose LGA as a spatial unit of analysis because of its clinical utility and its usefulness for public health interventions. In Victoria SA2 has a total of 523 regions and LGAs a total of 80. Yes, it is possible to find estimates of remoteness and socioeconomic status at a smaller geography like SA2 but in our scenario there is potential of identifiability issues and many SA2 regions will not have cases (outcome variable) and poses challenges in the analysis. We will consider the SA2 for future work.

Minor remark: P8, figure 2: I would suggest using a diverging colour scheme (different colours for high and low ends of the RR spectrum) to better delineate regions of low and high RR at a glance.

Author’s response: We agree with your suggestion, and we changed to a diverging colour scheme for better understanding of high and low ends of the RR spectrum (see the revised manuscript figure 2, P12).

Reviewer 2 Report

Comments and Suggestions for Authors

This study used Bayesian spatio-temporal multilevel modelling to examine patient-reported quality of life following prostate cancer surgery. The study has a certain value for public health practice but some issue need to be revised before being accepted for publication.

1. Line 121-122: Patient level data included: age (<55,56-65, 66-75, & 75-85). 75-85 should be 76-85. “<55” should be “<= 55”?

2. Line 129-130: The SEIFA (Socio-Economic Index for Area) has four variables (IRSAD, IRSD, IEO, IER) by Australian Bureau of Statistics. The authors need to explain the reason of using IRSD instead of other three variables.

3. Line 143-145: Does it mean postcode area corresponding to LGA? Does each postcode area belong to particular LGA, not two or more LGAs?

4. Line 192: GeoDA 1.22 version?

5. Table 1 and other tables, name of variables should be moved to the top line of each category, e.g., Age group should be in the same line with <55.

6. Table 2 and other places of the manuscript: p-value should be P-value as capital letter and italic format.

7. Line 223: “… patients in Victoria from 2015 to” should be: 2015 to 2021.

8. Line 367: “% change”?

9. Reference: the style of journals’ names should be unified. E.g., all should use full form like “New England Journal of Medicine”, or use abbreviated form like “Ca Cancer J Clin.” but not mixed format. The first letter of content words should be as capital letter. E.g., reference 3: “European urology oncology” should be “European Urology Oncology”.

Comments on the Quality of English Language

The manuscript need some language correction before being accepted for publication.

Author Response

Reviewer #2

This study used Bayesian spatio-temporal multilevel modelling to examine patient-reported quality of life following prostate cancer surgery. The study has a certain value for public health practice, but some issue needs to be revised before being accepted for publication.

  1. Line 121-122: Patient level data included: age (<55,56-65, 66-75, & 75-85). 75-85 should be 76-85. “<55” should be “<= 55”?

Author’s response: Thank you very much for your comment. We have corrected it in the revised manuscript, line 163.

  1. Line 129-130: The SEIFA (Socio-Economic Index for Area) has four variables (IRSAD, IRSD, IEO, IER) by Australian Bureau of Statistics. The authors need to explain the reason for using IRSD instead of the other three variables.

Author’s response: We thank the reviewer for the comment on IRSD and the other SEIFA indexes. SEIFA, developed by the ABS consists of the following four indexes:

The Index of Relative Socio-Economic Disadvantage (IRSD): is a general socio-economic index that summarises a range of information about the economic and social conditions of people and households within an area. IRSD only includes measures of relative disadvantage. A low score indicates a relatively greater disadvantage. This is our primary variable of interest and the hypothesis was to see the effect of IRSD on poor Quality of life among prostate cancer surgery patients. When we reviewed the literature, (https://bmccancer.biomedcentral.com/articles/10.1186/s12885-022-09389-4 , https://pubmed.ncbi.nlm.nih.gov/33862413/  ), we found IRSD to be a significant factor. We have explained this in the manuscript (see revised manuscript line 171).  

The Index of Relative Socio-Economic Advantage and Disadvantage (IRSAD): summarises information about the economic and social conditions of people and households within an area. This index includes both relative advantage and disadvantage measures. A low score indicates relatively greater disadvantage and a lack of advantage in general. This was not our interest of research hypothesis.

The Index of Education and Occupation (IEO): reflects the educational and occupational level of communities. There are no literature that support IEO as a factor for QoL.

The Index of Economic Resources (IER): focuses on the financial aspects of relative socio-economic advantage and disadvantage, by summarising variables related to income and housing. There are no literature that support IER as a factor for QoL.

  1. Line 143-145: Does it mean postcode area corresponding to LGA? Does each postcode area belong to particular LGA, not two or more LGAs?

Author’s response: Thank very much for this question regarding postcode and LGA concordance. In Victoria there are a total of 698 postcodes and 80 LGAs. A total of 318 postcode do not directly match to a specific LGA (means assigned in more than one LGA). Australia bureau of statistics? have calculated the area percentage of duplicated postcodes in each assigned LGA and then we assigned this postcode to the particular LGA with a greater area percentage share. For example, if one postcode covers 30% of its area in one LGA and 40% of its area in another LGA, we assigned this postcode in area of 40% coverage LGA. Finally, we got a total of 698 unique postcodes and aggregated them to LGA.   

  1. Line 192: GeoDA 1.22 version?

Author’s response: Thank you for your comment. Yes, it was GeoDA 1.22 version. We have corrected it in the revised manuscript (line 235).

  1. Table 1 and other tables, name of variables should be moved to the top line of each category, e.g., Age group should be in the same line with <55.

Author’s response: We agree with your comment and have corrected it accordingly (see the revised manuscript of all tables)

  1. Table 2 and other places of the manuscript: p-value should be P-value as capital letter and italic format.

Author’s response: Thank you for your comment.  We have corrected it accordingly (see revised manuscript Table 2).

  1. Line 223: “… patients in Victoria from 2015 to” should be: 2015 to 2021.

Author’s response: We have corrected it in the revised manuscript thank you for your comment (Line 241).

  1. Line 367: “% change”?

Author’s response: Yes it is % change. We fitted two models using Queen contiguity VS distance based spatial weight matrices. Our finding showed that there is a considerable % change in the estimate when we employ different adjacencies (with Queen contiguity as the reference method).

  1. Reference: the style of journals’ names should be unified. E.g., all should use full form like “New England Journal of Medicine” or use abbreviated form like “Ca Cancer J Clin.” but not mixed format. The first letter of content words should be as capital letter. E.g., reference 3: “European urology oncology” should be “European Urology Oncology”.

Author’s response: Thank you very much for your comment. We used Endnote version 20 software for referencing. This software is generating reference automatically at the end of the word document when we insert it. We appreciate your critical review of our manuscript. We corrected it in the revised manuscript based on your suggestion.

Reviewer 3 Report

Comments and Suggestions for Authors

1.   I suggest avoiding the use of abbreviations in the Abstract.

2.    On page 2, line 45, add " that " after "It is estimated…"

3. Under sub-section 3.6, "Model Convergence", I suggest adding model diagnostic and convergence plots in the appendix for a better understanding by the readers.

4.    Please check for any extra spaces throughout the manuscript.

5. About Novelty of Statistical Methods: A similar model was applied in a previously published study by the authors, and the current model is applied with minor changes. [https://pubmed.ncbi.nlm.nih.gov/33862413/]

Author Response

Reviewer #3

  1. I suggest avoiding the use of abbreviations in the Abstract.

Author’s response: Thanks for the comment. We agree with your suggestion and have avoided abbreviations in the Abstract

  1. On page 2, line 45, add " that " after "It is estimated…"

Author’s response: We have corrected it in the revised manuscript, thank you for your comment (line 62).

  1. Under sub-section 3.6, "Model Convergence", I suggest adding model diagnostic and convergence plots in the appendix for a better understanding by the readers.

Author’s response: Thank you for the comment. We have included the model diagnostic and convergence plots as supplementary file 3 in the revised submission.

  1. Please check for any extra spaces throughout the manuscript.

Author’s response: Thanks for your comment. We have gone through all the documents, and we have avoided extra spaces in the revised manuscript (see the revised manuscript).

  1. About Novelty of Statistical Methods: A similar model was applied in a previously published study by the authors, and the current model is applied with minor changes. [https://pubmed.ncbi.nlm.nih.gov/33862413/]

Author’s response: We thank the reviewer for your comment on the statistical novelty of our manuscript. The novelty of this study lies in the clinical application. There is no previous study that investigated poor quality of life among prostate cancer surgery patients in Victoria. We believe this study will help healthcare panners and policymakers to design effective geographic interventions to reduce variation in Quality of life at the LGA level.

Round 2

Reviewer 2 Report

Comments and Suggestions for Authors

The authors have revised the manuscript according to the reviewer's somments.

Comments on the Quality of English Language

Need minor revision.